# Evaluation of Polypropylene Reusability Using a Simple Mechanical Model Derived from Injection-Molded Products

**DOI:** 10.3390/polym17152107

**Published:** 2025-07-31

**Authors:** Tetsuo Takayama, Rikuto Takahashi, Nao Konno, Noriyuki Sato

**Affiliations:** 1Graduate School of Organic Materials Science, Yamagata University, Yamagata 990-8510, Japan; 2Department of Polymeric and Organic Materials Engineering, Yamagata University, Yamagata 990-8510, Japan; 3Industrial Technology Institute, Miyagi Prefectural Government, Miyagi 981-3206, Japan; konno-na794@pref.miyagi.lg.jp (N.K.); sato-no964@pref.miyagi.lg.jp (N.S.)

**Keywords:** isotacticity, mechanical properties, Poisson’s ratio

## Abstract

In response to growing global concerns about plastic waste, the development of efficient recycling technologies for thermoplastics has become increasingly important. Polypropylene (PP), a widely used commodity resin, is of particular interest because of the urgent need to establish sustainable material circulation. However, conventional mechanical property evaluations of injection-molded products typically require dedicated specimens, which involve additional material and energy costs. As described herein, we propose a simplified mechanical model to derive Poisson’s ratio and critical expansion stress directly from standard uniaxial tensile tests of molded thermoplastics. The method based on the true stress–true strain relationship in the small deformation region was validated using various thermoplastics (PP, POM, PC, and ABS), with results showing good agreement with those of the existing literature. The model was applied further to assess changes in mechanical properties of Homo-PP and Block-PP subjected to repeated extrusion. Both materials exhibited reductions in elastic modulus and critical expansion stress with increasing extrusion cycles, whereas Block-PP showed a slower degradation rate because of thermo-crosslinking in its ethylene–propylene rubber (EPR) phase. DSC and chemiluminescence analyses suggested changes in stereoregularity and radical formation as key factors. This method offers a practical approach for evaluating recycled PP and contributes to high-quality recycling and material design.

## 1. Introduction

Thermoplastic polymers (TPs) such as polypropylene (PP) are used extensively in widely diverse applications including packaging, automotive parts, and household products, primarily because of their low density, processability, and cost efficiency [1]. However, of the more than 400 million tons of plastic produced globally in 2022, only a small fraction was recycled effectively [2,3]. In fact, most post-consumer plastic waste is eventually incinerated or landfilled, exacerbating environmental concerns such as marine pollution and greenhouse gas emissions. Therefore, the development of practical recycling technologies, particularly for widely used resins such as PP, is urgently needed to promote a circular economy and to mitigate environmental impact. Specifically, with regard to TP there is a worldwide demand to promote mechanical recycling, which offers excellent resource recyclability. However, research in this area has not progressed sufficiently due to the absence of clear mechanical indicators of recyclability.

Mechanical properties of TP products, even when the same base material is used, can vary considerably depending on their associated molding conditions and geometries. Therefore, appropriate property evaluation for each product is fundamentally important [4,5]. Traditionally, mechanical testing involves the fabrication of dedicated specimens, which incurs high material, time, and energy costs. To promote resource-efficient and energy-efficient evaluation methods, simplified and practical testing approaches are increasingly necessary.

Uniaxial tensile testing is commonly used to assess the mechanical properties of molded TPs, providing valuable measurements such as the elastic modulus, yield stress, and elongation at break. Several models exist for the elastic and yielding behavior, but no comprehensive mechanical model for elongation at break has yet been adequately established [4]. Takayama et al. proposed a model to derive the critical expansion stress from elongation at break, analogizing the tensile fracture of TPs to that of metals [6]. This model incorporates the assumption that a triaxial stress state develops inside the specimen during uniaxial tension, leading to void initiation, growth, and eventual unstable fracture. The stress at which voids begin to grow unstably is defined as the critical expansion stress: a useful parameter to analyze material behavior. Because TPs typically have lower elastic moduli than those of metals, greater degrees of strain are required to reach comparable stress levels, making accurate determination of both the elastic modulus and Poisson’s ratio necessary for the model’s application.

Poisson’s ratio is typically measured using strain gauges or non-contact techniques such as digital image correlation (DIC) [7,8]. Unfortunately, strain gauges can introduce stiffness artifacts and have low signal-to-noise ratios in the small-strain region. Moreover, DIC accuracy depends heavily on image quality, consequently limiting its applicability to small-scale specimens. Takayama et al. proposed a method for predicting Poisson’s ratio using differential scanning calorimetry (DSC) [9] and a method for evaluating Poisson’s ratio using data from a three-point bending test [5]. Although Poisson’s ratio prediction by DSC can be performed using only a small amount of sample, it is time-consuming for evaluation and analysis. For Poisson’s ratio evaluation based on three-point bending test data, the area which can be evaluated is limited to the surface area of the molded product. Therefore, constructing a mechanical model to estimate Poisson’s ratio directly from tensile test data persists as an important challenge for understanding TP fracture behavior.

Mechanical recycling, a major recycling method for TPs, involves the reprocessing of waste plastics through repeated melt extrusion and injection molding [10,11]. Uzosike et al. reviewed small-scale mechanical recycling of PET, PE, and PP products, highlighting their low cost and energy efficiency, while also noting shortcomings such as contamination and deterioration caused by inadequate cleaning [11]. Because thermal and shear stresses during extrusion and molding can engender molecular weight reduction, the degradation of physical properties becomes a major concern [12,13]. Accordingly, quantitative evaluation of material properties after repeated extrusion is necessary for assessing mechanical recyclability. For example, in an investigation of repeated extrusion, Von Vacano et al. reported that repeated extrusion of polypropylene lowers melt viscosity and that this is due to a decrease in molecular weight [12]. Rust et al. reported that the weight-average molecular weight decreased from approximately 400,000 to 300,000 after nine extrusion cycles. They also reported that this decrease in molecular weight was a factor in the reduction in tensile strength. However, the change in molecular weight in this range is only in the high-molecular-weight region. It is therefore unlikely that this alone would reduce tensile strength [13]. In other words, the change in mechanical properties is not only due to the decrease in molecular weight, but also to other factors.

Polypropylene (PP) is a representative TP used extensively in automotive parts, packaging, and household appliances. Establishing recycling methods for PP is of pressing importance from a global resource sustainability perspective [14]. Joachim et al. reported that additives improved the phase structure and tensile impact strength of recycled LDPE, HDPE, and PP scraps [15]. Recycled PP has also been melt-compounded with waste printed circuit boards to create thermoplastic composites [16]. However, all of these approaches are based on compounding, so it is hard to call them true recycling methods. Ideally, we would achieve “closed-loop recycling,” where PP is reused for the same application. To increase PP’s reusability as a single material, we need to develop horizontal recycling technology. For this, we must understand how physical properties degrade due to repeated extrusion.

This study investigated the following two objectives. First, it aimed to establish a method to determine Poisson’s ratio based on tensile tests and derive critical expansion stress. Second, it sought to clarify the mechanical recyclability of polypropylene (PP) using critical expansion stress and elucidate its degradation factors. A mechanical model was first developed to estimate Poisson’s ratio from the elastic region of nominal stress–true strain curves. Its validity was confirmed across multiple resin types and molding temperatures. Subsequently, uniaxial tensile tests were conducted of PP samples with different extrusion histories to assess changes in the elastic modulus and Poisson’s ratio. The obtained parameters were used to calculate the critical expansion stress. Then the effects of repeated extrusion on material degradation were analyzed quantitatively. The aims of this study are to elucidate the mechanisms of property changes that occur during mechanical recycling of PP and to contribute to the development of practical recyclability evaluation methods.

## 2. Materials and Methods

### 2.1. Materials

The thermoplastic resins used for evaluating Poisson’s ratio in this study are presented in Table 1. The materials employed were the following: polypropylene (PP, Novatec MA1B; Japan Polypropylene Co., Ltd., Tokyo, Japan), polyoxymethylene (POM, Tenac 3010; Asahi Kasei Corp., Tokyo, Japan), acrylonitrile–butadiene–styrene copolymer (ABS, Kralastic GA-101; Nippon A&L Inc., Osaka, Japan), and polycarbonate (PC, Iupilon H-3000; Mitsubishi Engineering-Plastics Corp., Tokyo, Japan).

For evaluation of critical expansion stress under repeated extrusion, polypropylenes of two types were used, a homopolymer-grade PP (Homo-PP, Novatec MA3; Japan Polypropylene Co., Ltd., Tokyo, Japan) and a block copolymer-grade PP (Block-PP, Novatec BC03AD; Japan Polypropylene Co., Ltd.), as shown in Table 2.

### 2.2. Extrusion Molding

A flow chart including the sample preparation process conducted in this study is shown in Figure 1. Repeated melt extrusion of Homo-PP and Block-PP was conducted using a co-rotating twin-screw extruder (TEM-26SX; Shibaura Machine Co., Ltd., Tokyo, Japan). The numbers of extrusion cycles were set as 0 (E-0), 1 (E-1), 3 (E-3), and 5 (E-5) and pellets were prepared for each condition. The screw length-to-diameter ratio (L/D) was 48. The screw rotation speed was maintained as 200 rpm. The temperature profile from the hopper side was set as 180/180/200/200/200/200 °C. The residence time in the barrel was approximately 1 min. The resin temperature at the die exit was approximately 210 °C.

The appearance of the obtained pellets is shown in Figure 2. In the case of Homo-PP, noticeable yellowing was observed after more than three extrusion cycles, whereas Block-PP exhibited slight discoloration after the fifth cycle.

### 2.3. Injection Molding

Using a microelectric injection molding machine (C, Mobile0813; Shinko Sellbic Co., Ltd., Tokyo, Japan), the pellets prepared above were injection-molded into dumbbell-shaped specimens. Figure 3 depicts the shape of the molded product. In this study, dumbbell-shaped specimens equivalent to 1/4 scale of the multipurpose dumbbell specimen recommended in ISO 527 were fabricated [17]. The molding conditions used for the evaluation of Poisson’s ratio and critical expansion stress are presented, respectively, in Table 3 and Table 4.

### 2.4. Evaluation of Poisson’s Ratio by Tensile Testing

Uniaxial tensile tests were conducted in accordance with ISO 527 [17] using the molded dumbbell specimens. The equipment was a compact universal mechanical testing machine (FSA-1KE-1000N-L; Imada Co., Ltd., Aichi, Japan). A wedge-shaped chuck was employed to eliminate compliance and misalignment of the testing machine and specimens to the greatest extent possible. The gauge length was set as 22 mm. The tensile speed was 10 mm/min. Load–displacement curves were obtained up to a displacement of approximately 5 mm. From these curves, nominal stress–true strain curves were derived. The initial slope was evaluated as the elastic modulus *E*. Poisson’s ratio was calculated using Equation (1), which equates the hydrostatic component of true stress to the volumetric expansion stress under linear elastic conditions as follows:(1)$$\upsilon =\frac{\sigma }{3E\varepsilon -\sigma }$$
where *E* stands for the elastic modulus, ε represents the true strain, and σ denotes the true stress. This approach is based on the assumption that, in the initial elastic regime, the material behaves isotropically and follows Hookean mechanics. Specifically, the stress distribution under uniaxial tension can be decomposed into extensional and volumetric components, allowing υ to be isolated as a function of the applied stress and strain. Hooke’s law in three dimensions is expressed in Equation (2):(2)$$\sigma_{x}=\frac{\upsilon E}{\left(1-2\upsilon \right)\left(1+\upsilon \right)}\left(\varepsilon_{x}+\varepsilon_{y}+\varepsilon_{z}\right)+\frac{E\varepsilon_{x}}{1+\upsilon }$$
where subscripts *x*, *y*, and *z* denote the respective directions. For this study, the x-direction is defined as the loading direction in the tensile test, whereas the *y*- and *z*-directions are orthogonal to *x*. Substituting the uniaxial stress conditions (Equation (3)) and the volumetric strain approximation for small deformations (Equation (4)) into Equation (2) yields Equation (5) presented below:(3)$$\sigma_{x}=\sigma ,\sigma_{y}=\sigma_{z}=0$$(4)$$\varepsilon_{x}=\varepsilon ,\varepsilon_{y}=\varepsilon_{z}=-\upsilon \varepsilon$$(5)$$\sigma =\frac{\upsilon E\varepsilon }{\left(1+\upsilon \right)}+\frac{E\varepsilon }{1+\upsilon }$$
where the first term represents the expansive component. The second term corresponds to the extensional component. Rearranging these terms leads to Equation (6), which is commonly used in uniaxial tensile tests.(6)σ=Eε

Hydrostatic stress σ_m_ is given by Equation (7).(7)$$\sigma_{m}=\frac{\sigma_{x}+\sigma_{y}+\sigma_{z}}{3}$$

Substituting Equation (3) into Equation (7) gives the hydrostatic stress generated during uniaxial tensile testing, as shown in Equation (8):(8)$$\sigma_{m}=\frac{\sigma }{3}$$

Assuming that the first term in Equation (5) is equal to Equation (8), then Equation (9) is obtained as shown below.(9)$$\frac{\sigma }{3}=\frac{\upsilon E\varepsilon }{\left(1+\upsilon \right)}$$

Solving Equation (9) for Poisson’s ratio yields Equation (1). It is noteworthy that this evaluation method is limited to the small strain region. Figure 4 presents an example of the calculated Poisson’s ratio and the true strain curve. Evaluation was performed for the range within which a linear relation was obtained, including the compressive stress generated during specimen fixation. Poisson’s ratio was calculated using Equation (1). Specifically, the area up to the maximum point of this curve was regarded as the measurable area. The maximum value was used as the representative value of the molded product. Although Poisson’s ratio is fundamentally a function of strain and although it tends to decrease concomitantly with increasing strain, the small deformation region used for this study minimizes measurement error. More than 10 specimens were tested per sample. The mean and standard deviation of the results were calculated.

### 2.5. Evaluation of Critical Expansion Stress

Critical expansion stress was evaluated based on the uniaxial tensile test results of repeatedly extruded Homo-PP and Block-PP. The evaluation followed the mechanical model proposed by Takayama [6]. During this test the gauge length was 22 mm. The tensile speed was 50 mm/min. The nominal stress–true strain curve was derived from the obtained load–displacement curve. The initial slope was evaluated as the initial elastic modulus *E*, the slope of the nominal stress–true strain curve in the strain range for which Poisson’s ratio υ and Poisson’s ratio were found using the method described above as *E*_υ_. The true strain at fracture was evaluated as the elongation at break ε_B_.

The critical expansion stress was calculated as follows: It was assumed that the material fractures when the expansion stress under tensile loading reaches a critical value. If the void ratio at fracture is f_0_, then the relation between expansion stress σ_v_ and expansion strain ε_v_ is given as Equation (10) below:(10)$$\sigma_{v}=\frac{2E_{\upsilon }}{3(1+\upsilon )}\left(\frac{f_{0}}{1-f_{0}}\right)\varepsilon_{v}$$

The void ratio f_0_ was calculated using Equation (11):(11)$$f_{0}={\left(1+\frac{E_{\upsilon }}{4\sigma_{y}(1+\upsilon )}\right)}^{-1}$$

Here, σ_y_ is the stress at yield initiation, as defined by Equation (12) which follows:(12)$$\sigma_{y}=\sqrt{3}\alpha \left(T_{inj}-T_{test}\right)E_{\upsilon }\cos\left(\tan^{-1}2\upsilon \right)$$
where α is the averaged linear expansion coefficient as calculated using Baker’s empirical law [18]. Also, *T*_inj_ denotes the injection temperature and *T*_test_ signifies the test temperature, which is 23 °C ± 2 °C as room temperature. Also, ε_v_ is defined by Equation (13) which is as follows:(13)$$\varepsilon_{v}=\varepsilon_{B}\left(1-2\upsilon_{0}\right)+\varepsilon_{B}^{2}\left(\upsilon_{0}^{2}-2\upsilon_{0}\right)+\upsilon_{0}^{2}\varepsilon_{B}^{3}$$
where ε_B_ is the strain at break and υ_0_ is the Poisson’s ratio at fracture, as given by Equation (14) which is as follows:(14)$$\upsilon_{0}=\varepsilon_{B}^{-1}\left\{1-\sqrt{\frac{1}{1+\varepsilon_{B}}}\right\}-D$$
where it is assumed that the material is nearly incompressible during plastic deformation. A correction factor D of 0.001 is introduced for the Poisson’s ratio under the incompressibility assumption. Each sample was tested at least 10 times. The mean and standard deviation of the results were calculated.

### 2.6. Differential Scanning Calorimetry (DSC)

The central portions of the dumbbell specimens molded from Homo-PP and Block-PP were cut and subjected to DSC measurement using a differential scanning calorimeter (DSC Q2000; TA Instruments—Waters LLC, New Castle, DE, USA). The measurement conditions were the following: equilibrium temperature of 40 °C, heating rate of 10 °C/min, and maximum temperature of 230 °C. From the resulting heat flow curves, the melting point and the melting enthalpy were found. Crystallinity *X_c_* was calculated using Equation (15) as follows:(15)$$X_{c}=\frac{\Delta H_{m}}{\Delta H}$$
where Δ*H*_m_ is the melting enthalpy and Δ*H* is the enthalpy of fusion for a fully crystalline material, taken as 207 J/g in this study [19]. Using the obtained crystallinity and melting point, the free volume *f* was calculated using Equation (16) as follows [9]:(16)$$f=0.025+0.00048\left(T_{inj}-\frac{2}{3}T_{m}\right)\left(1-X_{c}\right)$$
where *T*_inj_ is the injection molding temperature and *T*_m_ is the melting point. This test was performed once for each sample.

### 2.7. Chemiluminescence Measurement

Chemiluminescence (CL) measurements were taken using pellets of repeatedly extruded Homo-PP and Block-PP with a high-temperature-compatible weak light emission detector (CLA FS3si; Tohoku Electronic Industrial Co., Ltd., Miyagi, Japan). The measurements were taken using a temperature-rising method. The temperature ranges were 50–140 °C for Homo-PP and 50–150 °C for Block-PP, with a heating rate of 10 °C/min and a measurement interval of 1 s. The light emissions were counted at each time point. The relation between the number of emissions and measurement time was determined.

## 3. Results

### 3.1. Validity of Poisson’s Ratio Determined from Tensile Testing

Figure 5 shows representative nominal stress–true strain curves obtained from tensile tests conducted for evaluating Poisson’s ratio. These curves are presented up to a true strain of 0.2. The curve for PP exhibited a peak at a true strain of approximately 0.07. PS reached its peak at around 0.03 and fractured at about 0.05. POM displayed no clear peak in its stress–strain behavior. PC showed a peak at a true strain of approximately 0.06. ABS reached a maximum near 0.03; it fractured within the range of 0.1–0.2.

Based on these curves, Poisson’s ratio at each true strain was calculated using the method described in Section 2.4. The resulting Poisson’s ratio–true strain curves are presented in Figure 6. For PP, the curve peaked at a true strain of approximately 0.01. For PS, Poisson’s ratio remained nearly constant between true strain values of 0.005 and 0.025. POM exhibited a peak at approximately 0.015, whereas PC and ABS, respectively, showed maxima at approximately 0.02 and 0.015–0.02. These peak points correspond to the linear region of the nominal stress–true strain curves shown in Figure 5, indicating that the proposed method is applicable within this deformation range.

Figure 7 and Table 5 show results obtained when comparing the Poisson’s ratios obtained in this study with values from the literature, where the numbers in parentheses in Table 5 represent standard deviations. The reference data were calculated using Takayama’s model based on bending strength and modulus from three-point bending tests [9]. The Poisson’s ratios obtained from this study show good agreement with values reported in the literature for all materials. The evaluation method proposed herein is judged to be valid.

The tendency of Poisson’s ratio to change with injection molding temperature was confirmed, and this change is closely related to the mobility of polymer molecular chains and the densification of amorphous regions. As reported by Takayama et al., the main factor is thought to be the increase or decrease in free volume formed in the molded body due to differences in injection conditions [9]. Injection molding is especially affected by changes in temperature conditions, which impact the viscosity and molecular orientation of the polymer in the molten state as well as densification of the molecules during cooling. For instance, high injection pressure or a fast-cooling rate restricts the movement of molecular chains, resulting in a relatively small free volume. It is inferred that this changes the volume elastic modulus and lateral deformation behavior of the material, thereby altering the Poisson’s ratio. Therefore, it is possible to optimize mechanical properties, including Poisson’s ratio, by controlling the injection conditions.

### 3.2. Extrusion Temperature Effects on the Critical Expansion Stress of Polypropylene

Figure 8 presents representative nominal stress–true strain curves obtained from tensile tests conducted for critical expansion stress evaluation under various extrusion temperatures. All conditions exhibited a clear yield phenomenon following the initial elastic region. After yielding, stress initially decreased with further deformation, transitioning into a stable plateau. It finally rose again beyond the true strain of approximately 1.0 before failure.

From the curves for Homo-PP in Figure 8a, the following trends were observed with increasing extrusion temperature. The initial peak stress remained nearly constant at 40–42 MPa for 180 °C, 200 °C, and 220 °C, but it dropped to approximately 34 MPa at 240 °C. In the strain-hardening region (true strain > 1.0), all samples showed stress recovery, although the slope at 240 °C was clearly more gradual. The fracture strain values were approximately 1.65 at 180 °C and 1.45 at both 200 °C and 220 °C, increasing to approximately 1.8 at 240 °C.

It is noteworthy that specimens extruded at 240 °C showed distinctly different mechanical behavior, with lower stress levels and more gradual strain hardening. These trends suggest that the higher extrusion temperature induces qualitative changes in the microstructure.

As presented in Figure 8b, the curves for Block-PP also showed decreasing initial peak stress from about 32 MPa at 180 °C to 27 MPa at 240 °C. Fracture strain increased from about 1.5 at 180 °C to about 1.7 at 240 °C, indicating improved ductility with rising extrusion temperature.

Figure 9 and Table 6 show the critical expansion stress as a function of extrusion temperature for Homo-PP and Block-PP, with error bars showing standard deviations. In all cases, Homo-PP exhibited markedly higher critical expansion stress (20–28 MPa) than that of Block-PP (14–19 MPa). For both materials, increasing the extrusion temperature led to a clear reduction in critical expansion stress, especially between 180 °C and 200 °C. For instance, Homo-PP’s stress dropped from 28 MPa at 180 °C to 22 MPa at temperatures higher than 220 °C, whereas Block-PP’s dropped from 19 MPa to 14 MPa over the same temperature range.

### 3.3. Repeated Extrusion Effects on the Critical Expansion Stress of Polypropylene

Figure 10 presents representative nominal stress–true strain curves for each extrusion cycle. In Figure 10a, Homo-PP shows notable mechanical degradation with increasing extrusion repetitions. The initial peak stress decreased from approximately 43 MPa (E-0) to 42 MPa (E-1), 38 MPa (E-3), and 35 MPa (E-5). Among these, E-3 exhibited the latest fracture (true strain ~1.5), whereas E-0, E-1, and E-5 fractured around 1.3. Near-identical behavior of E-0 and E-1 suggests that a single extrusion does not strongly affect the mechanical properties, but repeated extrusions likely cause chain scission and embrittlement.

In Figure 10b, Block-PP samples also showed decreasing mechanical strengths with repeated extrusions. The initial peak stress dropped from 30–32 MPa (E-0 and E-1) to ~30 MPa (E-3) and ~26 MPa (E-5). Strain hardening above true strain ~1.3 occurred similarly across conditions, but the fracture strain was slightly higher for E-5. These results suggest that whereas a single extrusion has only a minimal effect, repeated cycles lead to gradual strength loss and embrittlement.

Figure 11 and Table 7 show the relationship between critical expansion stress and extrusion cycle number, with error bars indicating standard deviation. For Homo-PP the critical expansion stress remained nearly constant at approximately 22 MPa for both 0 and 1 extrusion cycles, indicating that a single extrusion cycle has only a minimal effect on mechanical properties. However, after three extrusion cycles the stress decreased considerably to approximately 16 MPa, and dropped further to approximately 13 MPa after five cycles. The error bars are larger at three extrusion cycles, but this greater error is attributable to the scattering of values closer to the results of one extrusion cycle and the results of five extrusion cycles. These findings indicate that Homo-PP is prone to marked mechanical degradation through repeated extrusion.

In contrast, Block-PP exhibited a critical expansion stress of approximately 18 MPa for both 0 and 1 extrusion cycles, with minimal variation, implying that it is more resistant to degradation. A slight decrease in stress was observed after three cycles, but the extent of variation was small. The error bars remained narrow. Even after five extrusion cycles, the stress decreased to around 12 MPa, but the overall degradation progressed more gradually than it did for Homo-PP.

These results demonstrate clearly that, compared to Homo-PP, Block-PP exhibits higher resistance to mechanical degradation from repeated extrusion. Whereas Homo-PP showed a sharp drop in performance beyond the third extrusion cycle, Block-PP maintained stable properties even with multiple reprocessing steps.

In summary, both Homo-PP and Block-PP exhibited mechanical degradation with repeated extrusion. However, their degradation rates differed as Homo-PP showed rapid strength loss, whereas Block-PP degraded more gradually. These findings indicate that structural changes occurring during extrusion vary by material type, providing valuable insights for material selection in recycling applications.

## 4. Discussion

### 4.1. Influence of Extrusion Temperature on the Critical Expansion Stress of Polypropylene

As shown in Section 3.2, both Homo-PP and Block-PP exhibited a common trend of decreased yield peak stress and increased fracture strain with rising extrusion temperature. This section presents detailed discussion of the underlying microstructural changes and the involvement of thermal chemical reactions, from the perspective of elastic properties and stereoregularity.

First, the temperature dependencies of the tensile modulus (*E*) and Poisson’s ratio (υ), shown in Figure 12, are reviewed. The results of Poisson’s ratio are presented in Figure 12a. Poisson’s ratios of both PP-based materials remained almost constant over the range of 180–240 °C. By contrast, the tensile modulus shown in Figure 12b decreased remarkably, concomitantly with increasing temperature. Specifically, Homo-PP remained stable at temperatures up to 220 °C, but dropped by approximately 200 MPa at 240 °C, whereas Block-PP showed a 200 MPa decrease by 200 °C and it then continued to decrease gradually.

Figure 13 and Table 8 present the DSC measurement results. Figure 13a,b show the first heating heat flow curves obtained by differential scanning calorimetry (DSC) measurements after processing homo-polypropylene (Homo-PP) and block-polypropylene (Block-PP) at different extrusion temperatures (180, 200, 220, and 240 °C), respectively. A distinct melting peak was observed around 150–170 °C for both materials, reflecting the melting behavior attributed to the crystalline phase of polypropylene.

As the extrusion temperature increased, the position of the melting peak (melting point) and the melting initiation temperature showed a slight tendency to decrease. However, there was no clear change in the area of the melting peak (enthalpy of melting), suggesting that crystallinity itself may not be significantly affected by thermal history despite minute changes in the crystalline structure due to thermal history. For Homo-PP in particular the sharpness of the melting peak decreased with increasing extrusion temperature, indicating impaired size distribution of the crystals and uniformity of crystallinity. Conversely, Block-PP’s melting peak spread was larger initially, likely due to the diverse crystal morphology resulting from the block copolymerization structure of ethylene and propylene. Additionally, Block-PP retains clear melting peaks at high extrusion temperatures (240 °C), indicating higher structural stability than Homo-PP against thermal history.

Figure 13c confirms that the crystallinity found for both materials showed minimal dependence on the extrusion temperature. Figure 13d shows the relation between the extrusion temperature and free volume for both Homo-PP and Block-PP. Across all temperature conditions, Homo-PP exhibited consistently higher free volume values than Block-PP, indicating looser molecular packing or greater chain mobility in its amorphous regions.

With increasing extrusion temperature, the free volume of Homo-PP showed a slight increasing trend, particularly at temperatures higher than 230 °C. This behavior presumably results from thermally induced radical formation and the consequent disruption of stereoregularity, which engenders expansion in the amorphous phase. By contrast, Block-PP exhibited a stable or slightly decreasing trend in free volume as the extrusion temperature increased. This suppression of the free volume increase is attributed to the presence of the ethylene–propylene rubber (EPR) phase. The thermally induced crosslinking reactions within the EPR phase are likely to consume the generated radicals, thereby restricting their propagation into the amorphous polymer matrix and thereby inhibiting the loosening of molecular packing [20].

These results suggest that, whereas Homo-PP is more susceptible to thermal structural relaxation and radical-induced degradation at elevated temperatures, Block-PP maintains a more stable structure because of the radical-scavenging effect of the EPR phase. This structural stability contributes to the superior thermal and mechanical durability of Block-PP under repeated thermal processing.

Therefore, the observed decrease in tensile modulus is regarded as deriving primarily from changes in stereoregularity and fluctuations in free volume within the amorphous phase, rather than from changes in crystallinity. Nomura et al. investigated the relationship between the stereoregularity of PP and the flexural modulus and crystallinity. They reported that the crystallinity decreases and the flexural modulus decreases when the stereoregularity is disrupted [21]. As shown by Takayama et al. [9], the polymer free volume correlates with the tensile modulus and Poisson’s ratio. Therefore, thermally induced increases in free volume might also contribute to a decrease in *E*. Since the formation of radicals induces molecular cleavage, the molecular weight decreases. However, this change occurs within the range of high molecular weight and is considered a minor factor that contributes to the decrease in mechanical properties. Von Vacano et al. and Rust et al. have also reported that a decrease in molecular weight decreases melt viscosity [12,13]. This implies that decreasing molecular weight increases molecular mobility, which should promote crystallization and increase crystallinity; however, the change in crystallinity was slight, suggesting that decreasing molecular weight with increasing extrusion temperature has a small effect.

Figure 14 presents result of chemiluminescence (CL) measurements. Iedema et al. [22] analyzed the relation between radical generation during thermal degradation and stereoregularity rearrangement, reporting that at high-temperatures radicals abstract hydrogen between amorphous chains, leading to chain scission and re-hydrogenation, thereby modifying the molecular alignment. Applying this mechanism to the present extrusion process, the increased luminescence counts observed with rising temperature indicate increased radical generation, suggesting enhanced disruption of stereoregularity. For Homo-PP the increased luminescence counts at 240 °C align with the marked decrease in *E*, thereby supporting the conclusion that thermal degradation and radical formation primarily drive the decline in elasticity.

In contrast, Block-PP exhibited fewer luminescence events than Homo-PP under the same temperature conditions. The decrease in *E* was more gradual. This difference is attributed to the radical scavenging mechanism of the ethylene–propylene rubber (EPR) phase included in Block-PP. EPR readily incorporates radicals through crosslinking, with the generated radicals being absorbed into the amorphous rubber phase, suppressing their propagation into the amorphous polymer phase. As a result, the *E* and υ of Block-PP showed less temperature dependence, contributing to a smaller variation in critical expansion stress with temperature.

### 4.2. Influence of Repeated Extrusion on the Critical Expansion Stress

As presented in Section 3.3, although the mechanical properties of both PP-based materials deteriorated with an increasing number of extrusion cycles, their rates of degradation differed considerably. The dependence of the tensile modulus (*E*) and Poisson’s ratio (υ) on the number of extrusions is reviewed in Figure 15. According to the figure, after five extrusion cycles, *E* decreased by approximately 400 MPa for Homo-PP and by 250 MPa for Block-PP. The Poisson’s ratio increased by about 0.05 for Homo-PP, whereas for Block-PP it remained nearly unchanged.

Figure 16 and Table 9 present results of the DSC measurements. Figure 16a,b show the DSC measurement results for polypropylene (PP) with different numbers of cyclic extrusion cycles (E-0 to E-5), corresponding to Homo-PP and Block-PP, respectively. In both samples, a distinct endothermic peak was observed at about 160–170 °C, reflecting thermal absorption during the melting of PP’s crystalline phase.

Generally, the top temperature of the melting peak decreased slightly as the number of extrusion cycles increased, and the peak shape gradually broadened. These changes suggest thermo-mechanical degradation of the polymer chain due to repeated extrusion. Furthermore, the area of the melting peak (enthalpy of melting) appears to have decreased slightly, suggesting a potential decrease in crystallinity. However, the baseline of each curve remains stable around the melting peak, confirming the reproducibility of the DSC measurements.

Figure 16a shows a single, relatively sharp melting peak for Homo-PP at each extrusion frequency, reflecting its homogeneous crystalline structure. As the number of extrusion cycles increased, the peak broadened slightly and the peak top temperature decreased gradually, suggesting that repeated extrusion affects the crystal size and integrity of Homo-PP.

Figure 16b shows a single distinct melting peak for Block-PP as well, but the peak width is slightly wider than that of the homopolymer. This may reflect the structural heterogeneity of a block copolymer. The decrease and broadening of the peak with increasing extrusion frequency indicates that the crystal structure of Block-PP also undergoes changes with repeated extrusion.

Although Homo-PP exhibited a decrease in crystallinity of about 0.03 after five cycles, Block-PP showed no noteworthy change, as shown in Figure 16c. This trend suggests that the cumulative application of thermal and shear energy promotes repeated radical generation, severely disrupting the amorphous phase structure of Homo-PP in particular.

Figure 16d portrays the relationship between the number of extrusion cycles and the free volume for Homo-PP and Block-PP. For both materials a gradual increase in free volume is observed with repeated extrusion, indicating progressive structural relaxation and chain rearrangement in the amorphous phase caused by thermal and shear energy input during reprocessing.

For Homo-PP the increase in free volume is more pronounced, especially after the third extrusion cycle. This trend suggests that repeated extrusion promotes radical formation and stereoregularity disruption, which in turn engenders increased molecular mobility and expansion of the amorphous regions. The steep increase in free volume reflects the susceptibility of Homo-PP to thermal degradation and cumulative damage during multiple processing steps.

By contrast, Block-PP shows a more moderate increase in free volume over the extrusion cycles. The presence of the ethylene–propylene rubber (EPR) phase is thought to play a key role in mitigating the accumulation of free volume. The EPR phase can consume radicals via thermal crosslinking reactions, thereby limiting the structural damage and preserving molecular order within the matrix. As a result, the growth in free volume is suppressed compared to Homo-PP.

These findings indicate that free volume can serve as a sensitive indicator of thermal and structural degradation in polypropylene during mechanical recycling, with Block-PP exhibiting superior resistance to degradation because of its rubber-modified structure.

Figure 17 shows the results obtained from chemiluminescence measurements. In Figure 17a, the luminescence counts during the heating process for Homo-PP increased considerably with the number of extrusion cycles, even more than during a single extrusion at 240 °C, clearly indicating the accumulation of damage.

Although luminescence counts also increased in Block-PP, as shown in Figure 17b, the increase was much more subdued than for Homo-PP under the same conditions. This finding suggests that the thermal crosslinking effect of the EPR phase consumed radicals and mitigated cumulative degradation. Therefore, the role of EPR during repeated extrusion is crucially important for maintaining the quality of recycled material. Optimizing the EPR content and controlling its degree of crosslinking are effective strategies for suppressing the degradation of mechanical properties.

In summary, the deterioration of mechanical properties in polypropylene is driven mainly by radical generation because of thermal degradation and because of the subsequent disruption of stereoregularity, leading to reduced yield stress and critical expansion stress. Findings also indicate that in Block-PP, which contains EPR, radical consumption via thermal crosslinking suppressed the progression of degradation. These findings underscore the importance of optimizing material composition and extrusion conditions in recycling applications.

Moreover, this study suggests that radical generation and stereoregularity can serve as new evaluation indices for assessing recycled polypropylene. Manago et al. proposed a method to evaluate the degree of degradation of polypropylene and polyethylene using terahertz spectroscopy [23]. Future studies are expected to lead to the development of technologies that can facilitate the measurement of these structural indicators in real time, such as terahertz spectroscopy, which can establish a method for sorting and monitoring high-quality recycled materials. Furthermore, this method may be applicable to the evaluation of the mechanical recyclability of other polymeric materials. In addition, considering the effects of steric regularity noted in this study may lead to the development of molding and processing technologies that enable horizontal recycling.

## 5. Conclusions

The Poisson ratio estimation method developed in this study, based on uniaxial tensile testing, produced values that closely matched values from the literature across various thermoplastic resins, confirming the validity and reliability of the approach. Furthermore, because this method provides a means of direct evaluation of Poisson’s ratio from molded products, it obviates specialized test specimens, enabling efficient characterization of mechanical properties.

As the extrusion temperature increased, both materials exhibited a decrease in yield stress and an increase in fracture strain, indicating enhanced ductility at higher temperatures. Regarding the tensile modulus, Homo-PP showed a marked decrease at 240 °C, whereas Block-PP exhibited a gradual decline at temperatures higher than 200 °C. Although DSC analysis was conducted to identify the cause, no marked change in crystallinity was observed. These reductions in tensile modulus are therefore attributed to a disruption in stereoregularity caused by thermally induced radical generation. Chemiluminescence measurements confirmed that higher extrusion temperatures led to increased radical generation. In Homo-PP this increased radical generation caused notable deterioration of stereoregularity, crystallinity, and mechanical properties. By contrast, for Block-PP the thermal crosslinking reaction of the EPR phase consumed radicals, thereby suppressing changes in tensile modulus and Poisson’s ratio and minimizing the temperature dependence of the material.

Evaluation of property degradation from repeated extrusion revealed that both materials experienced decreased modulus and strength, although the extent differed considerably. In Homo-PP a marked reduction in tensile modulus was observed even after the first extrusion cycle. After five cycles the modulus had dropped by approximately 400 MPa compared to the initial state. By contrast, Block-PP showed a more gradual decline because of the EPR-phase consumption of radicals and the formation of crosslinked structures, thereby suppressing degradation even after multiple extrusions. These findings demonstrate that recycling resistance differs considerably depending on a material’s composition, with Block-PP exhibiting superior resistance to degradation.

The critical expansion stress, as calculated from the tensile properties, consistently decreased concomitantly with increasing extrusion cycles, confirming its effectiveness as a quantitative indicator of material degradation because of repeated extrusion.

Overall, although both Homo-PP and Block-PP showed degradation with re-extrusion, the severity differed markedly. Block-PP exhibited higher recyclability because of radical consumption and degradation suppression by the EPR phase, whereas Homo-PP was prone to early-stage strength loss. These insights suggest that structural indicators such as radical generation and stereoregularity can serve as effective metrics for evaluating and selecting high-quality recycled materials from waste polypropylene.

## Figures and Tables

**Figure 1 polymers-17-02107-f001:**
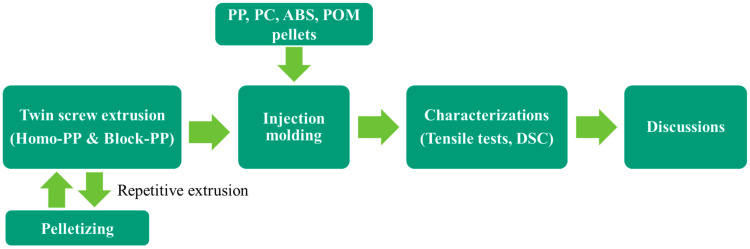
Flow chart conducted in this study.

**Figure 2 polymers-17-02107-f002:**
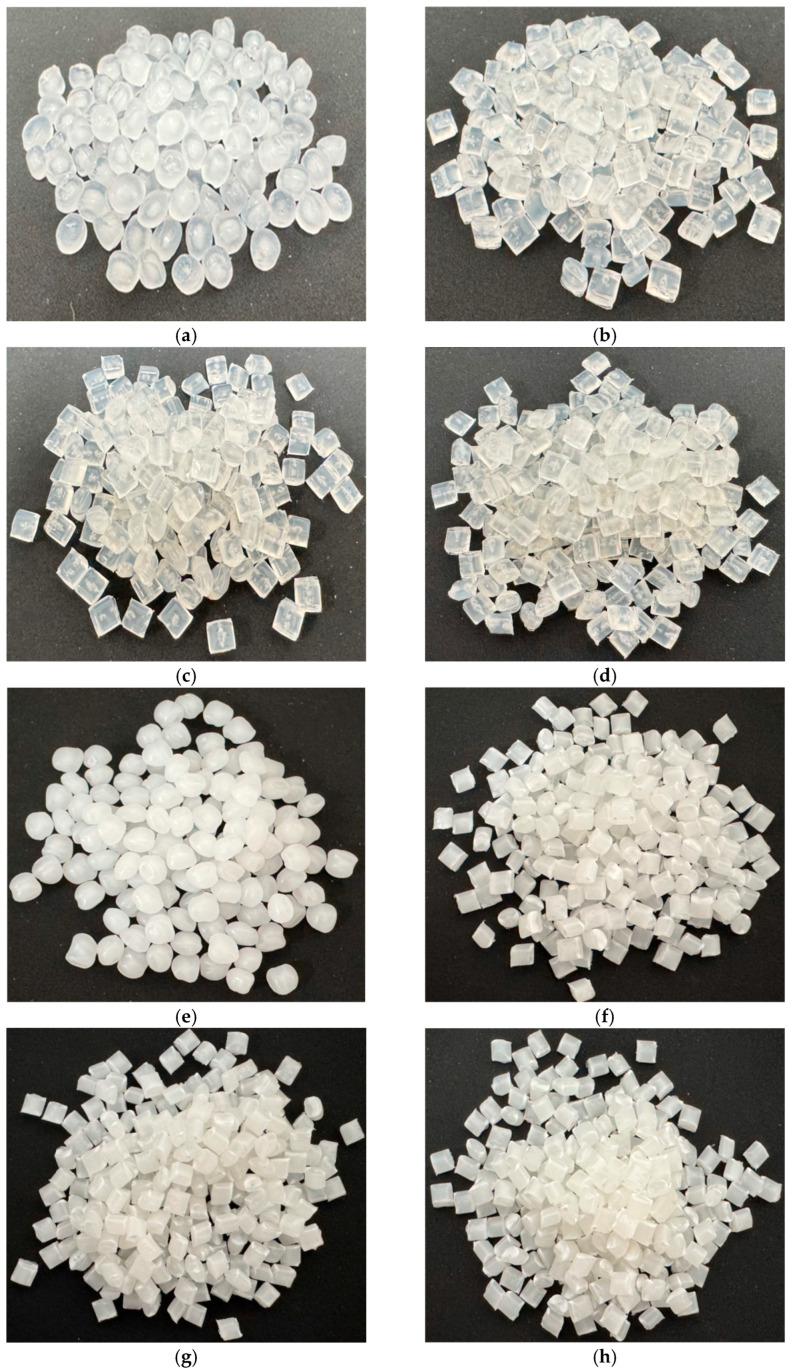
Pellets obtained from repeated extrusion. Pellets are approximately 3 mm: (**a**) Homo-PP E-0, (**b**) Homo-PP E-1, (**c**) Homo-PP E-3, (**d**) Homo-PP E-5, (**e**) Block-PP E-0, (**f**) Block-PP E-1, (**g**) Block-PP E-3, and (**h**) Block-PP E-5.

**Figure 3 polymers-17-02107-f003:**
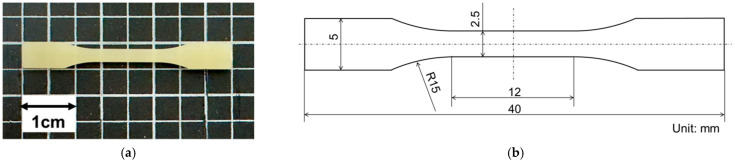
(**a**) Appearance and (**b**) geometry of dumbbell-shaped specimen. Thickness is 1 mm.

**Figure 4 polymers-17-02107-f004:**
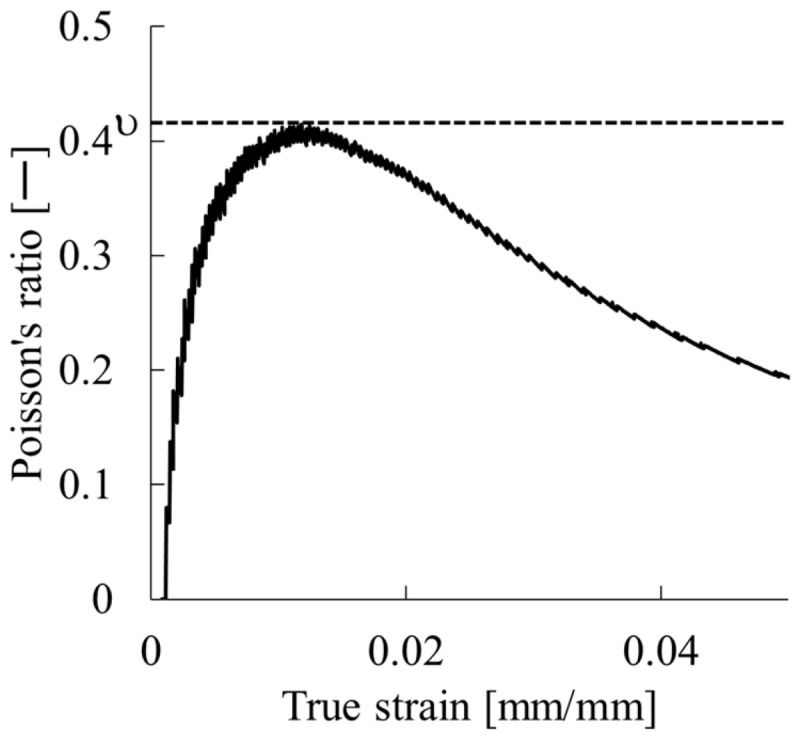
Example of Poisson’s ratio determination from the Poisson’s ratio–true strain curve.

**Figure 5 polymers-17-02107-f005:**
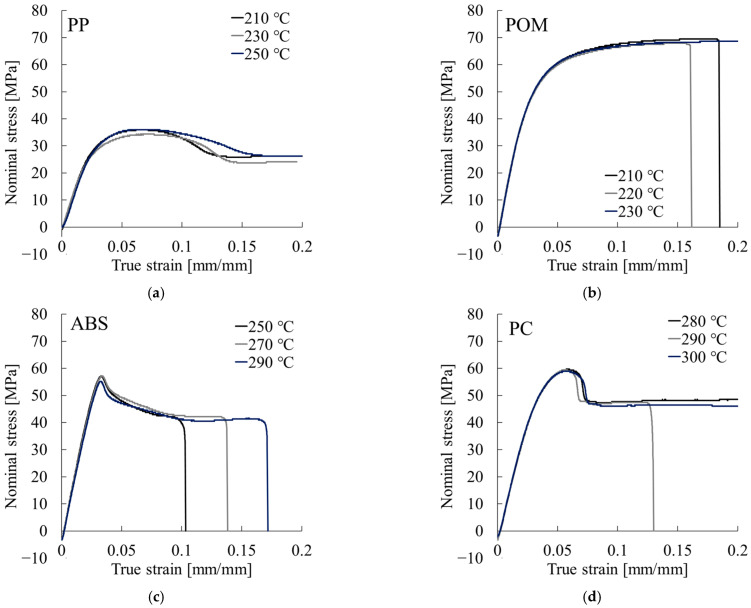
Nominal stress–true strain curves obtained for evaluation of Poisson’s ratio: (**a**) PP, (**b**) POM, (**c**) ABS, and (**d**) PC.

**Figure 6 polymers-17-02107-f006:**
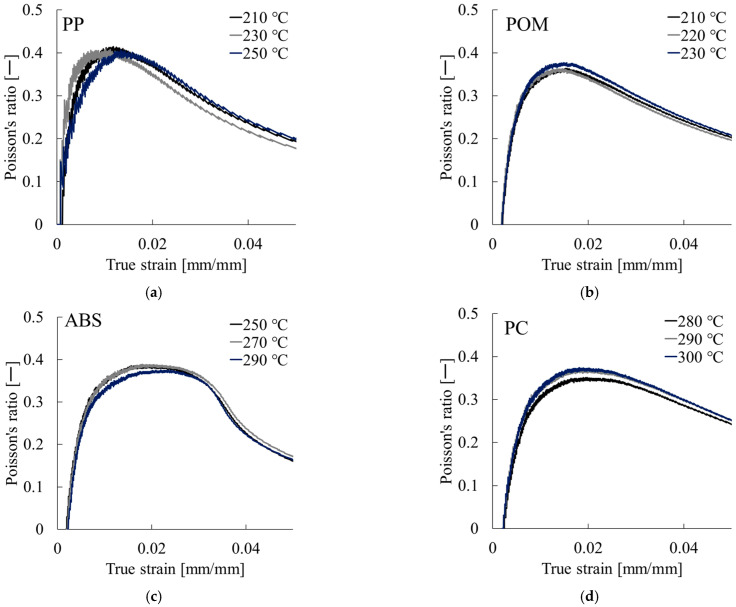
Examples of Poisson’s ratio–true strain curves: (**a**) PP, (**b**) POM, (**c**) ABS, and (**d**) PC.

**Figure 7 polymers-17-02107-f007:**
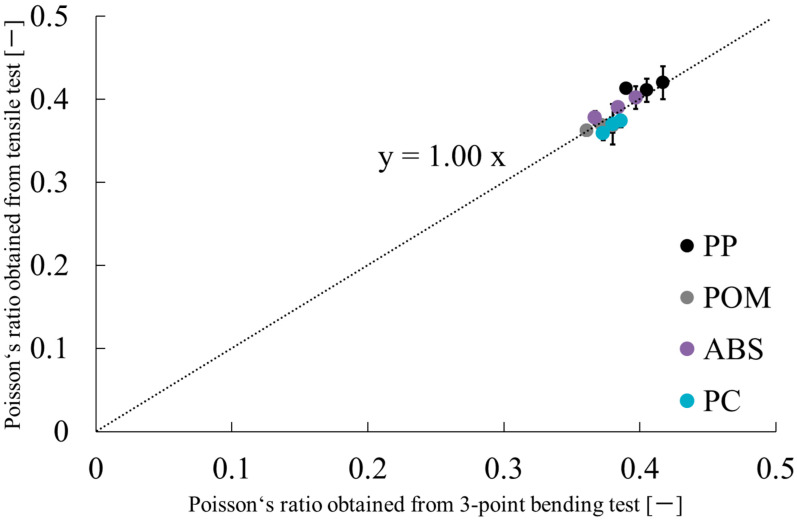
Experimentally obtained results of Poisson’s ratio obtained from this study and values obtained from the literature [9].

**Figure 8 polymers-17-02107-f008:**
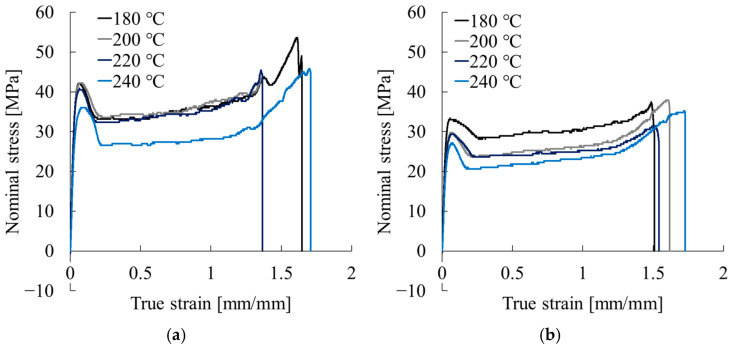
Nominal stress–true strain curves obtained from tensile tests conducted for critical expansion stress evaluation under various extrusion temperatures: (**a**) Homo-PP and (**b**) Block-PP.

**Figure 9 polymers-17-02107-f009:**
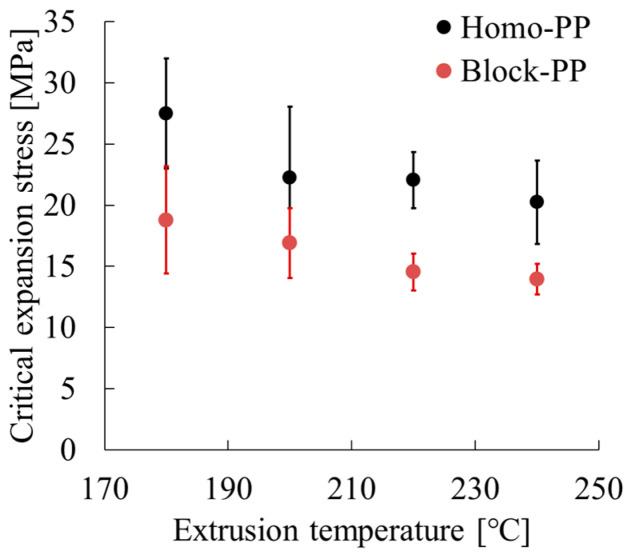
Critical expansion stress as a function of extrusion temperature.

**Figure 10 polymers-17-02107-f010:**
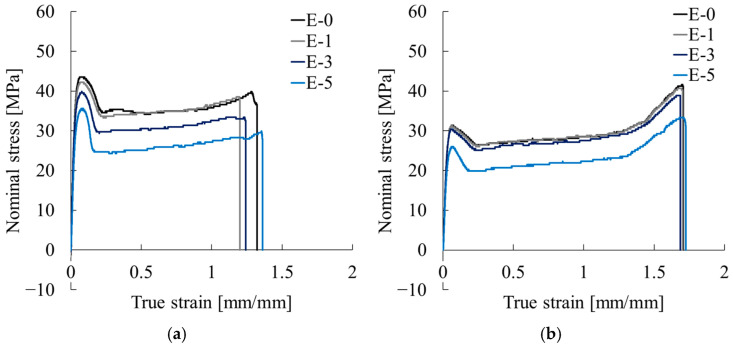
Nominal stress–true strain curves obtained from tensile tests conducted for critical expansion stress evaluation under various extrusion cycles: (**a**) Homo-PP and (**b**) Block-PP.

**Figure 11 polymers-17-02107-f011:**
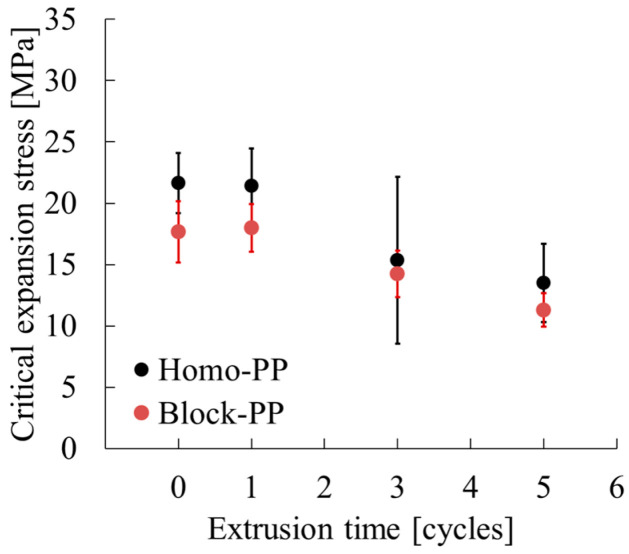
Critical expansion stress as a function of extrusion cycles.

**Figure 12 polymers-17-02107-f012:**
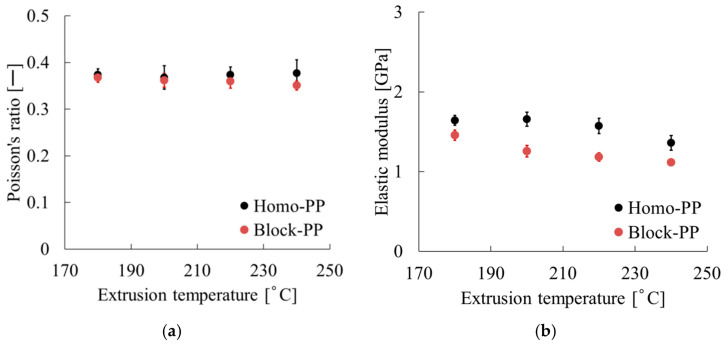
Extrusion temperature dependence of (**a**) Poisson’s ratio and (**b**) tensile modulus.

**Figure 13 polymers-17-02107-f013:**
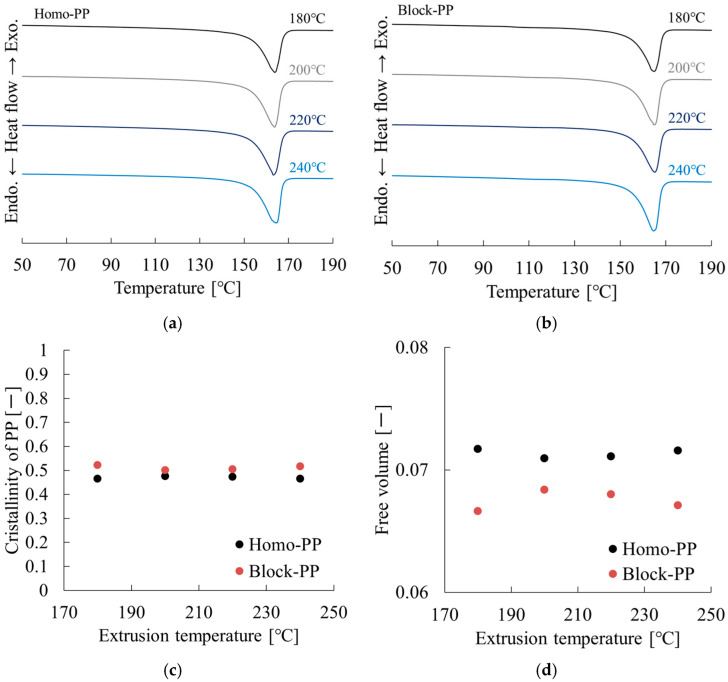
Extrusion temperature dependences of DSC curves of (**a**) Homo-PP and (**b**) Block-PP, (**c**) crystallinity, and (**d**) free volume of Homo-PP and Block-PP.

**Figure 14 polymers-17-02107-f014:**
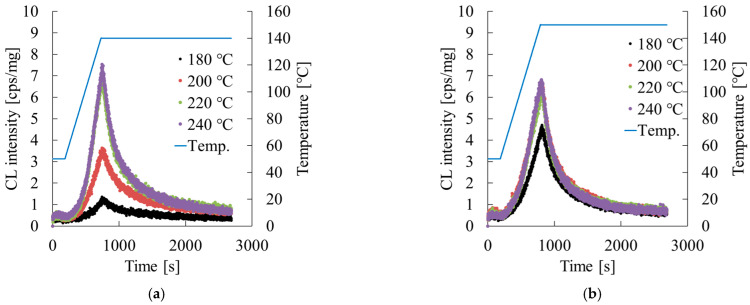
Extrusion temperature dependence of chemiluminescence (CL) measurements of (**a**) Homo-PP and (**b**) Block-PP.

**Figure 15 polymers-17-02107-f015:**
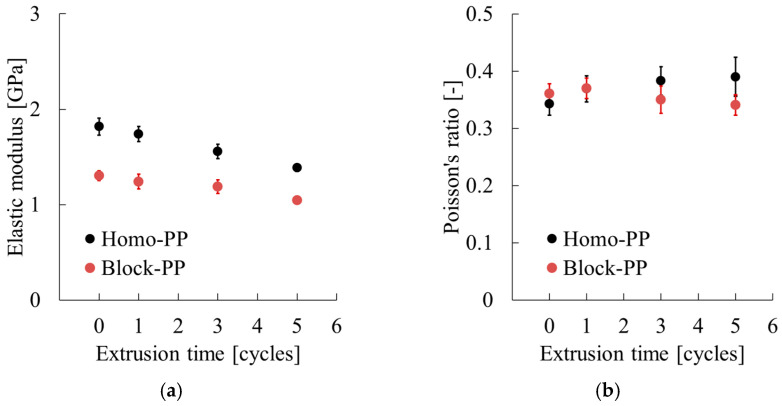
Extrusion cycle dependence of (**a**) Poisson’s ratio and (**b**) tensile modulus.

**Figure 16 polymers-17-02107-f016:**
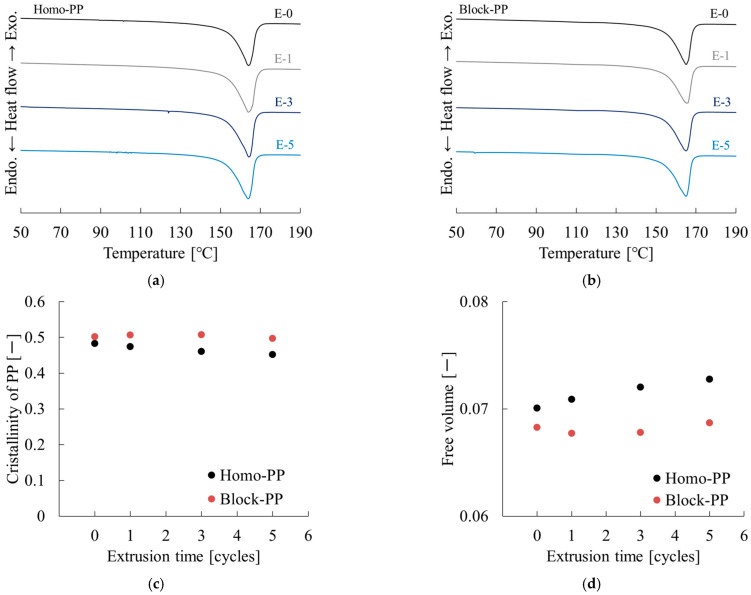
Extrusion cycle dependences of DSC curves of (**a**) Homo-PP and (**b**) Block-PP, (**c**) crystallinity, and (**d**) free volume of Homo-PP and Block-PP.

**Figure 17 polymers-17-02107-f017:**
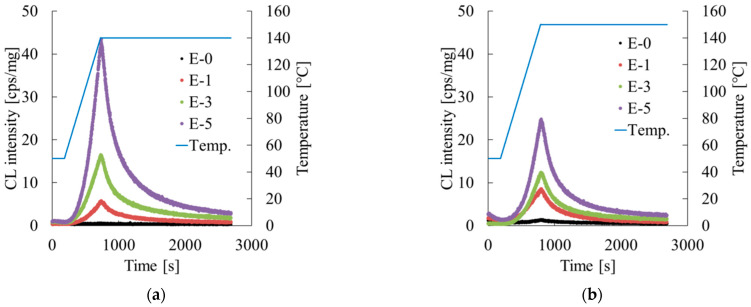
Extrusion cycle dependence of chemiluminescence (CL) measurements of (**a**) Homo-PP and (**b**) Block-PP.

**Table 1 polymers-17-02107-t001:** Material information for evaluating Poisson’s ratio.

Material	Code	Manufacturer	Grade
Polypropylene	PP	Japan Polypropylene Co., Ltd., Tokyo, Japan	Novatec-PP MA1B
Polyoxymethylene	POM	Asahi Kasei Corp., Tokyo, Japan	TENAC 3010
ABS copolymer	ABS	Nippon A&L Inc., Osaka, Japan	Kralastic GA-101
Polycarbonate	PC	Mitsubishi Engineering-Plastics Corp., Tokyo, Japan	Iupilon H-3000

**Table 2 polymers-17-02107-t002:** Material information for evaluating critical expansion stress under repeated extrusion.

Material	Code	Manufacturer	Grade
Homo-type Polypropylene	Homo-PP	Japan Polypropylene Co., Ltd., Tokyo, Japan	Novatec-PP MA3
Block-type Polypropylene	Block-PP	Japan Polypropylene Co., Ltd., Tokyo, Japan	Novatec-PP BC03AD

**Table 3 polymers-17-02107-t003:** Injection molding conditions for evaluating Poisson’s ratio.

Code	*T*_inj_ [°C]	*T*_mold_ [°C]	*V*_inj_ [m/s]	*P*_hold_ [MPa]	*T*_inj_ [s]	*T*_cool_ [s]
PP	210/230/250	50	10	42	45	15
POM	200/210/220	100	30	70	15	10
ABS	250/270/290	50	30	49	15	10
PC	280/290/300	100	30	11	30	10

**Table 4 polymers-17-02107-t004:** Injection molding conditions for evaluating critical expansion stress.

Code	*T*_inj_ [°C]	*T*_mold_ [°C]	*V*_inj_ [m/s]	*P*_hold_ [MPa]	*T*_inj_ [s]	*T*_cool_ [s]
Homo-PP	200	50	10	56	45	15
Block-PP	200	50	10	49	45	15

**Table 5 polymers-17-02107-t005:** Poisson’s ratio obtained from tensile tests and from the relevant literature [9].

Code	*T*_inj_ [°C]	*E* [MPa]	Poisson’s Ratio [—]	Poisson’s Ratio [9] [—]
PP	210	1491 (37)	0.413 (0.003)	0.390
	230	1433 (68)	0.411 (0.014)	0.405
	250	1375 (56)	0.420 (0.020)	0.417
POM	200	2498 (16)	0.362 (0.002)	0.361
	210	2537 (26)	0.369 (0.006)	0.372
	220	2417 (49)	0.368 (0.008)	0.380
ABS	250	2400 (22)	0.378 (0.008)	0.367
	270	2376 (14)	0.390 (0.001)	0.384
	290	2352 (5)	0.402 (0.013)	0.397
PC	280	2078 (12)	0.359 (0.008)	0.373
	290	2011 (23)	0.370 (0.024)	0.380
	300	1994 (29)	0.374 (0.007)	0.386

**Table 6 polymers-17-02107-t006:** Parameters used to determine critical expansion stress at different extrusion temperatures. Here, σ_y_ represents the stress at yield initiation as calculated using Equation (12), which is not the initial peak stress.

Code	Extrusion Temp. [°C]	*E* [MPa]	*E*_υ_ [MPa]	υ [—]	ε_B_ [—]	σ_y_ [MPa]	*f*_0_ [—]	σ_v_ [MPa]
Homo-PP	180	1642 (63)	1142 (59)	0.374 (0.013)	1.63 (0.01)	23.9 (1.1)	0.103 (0.005)	27.5 (4.5)
	200	1658 (89)	1191 (111)	0.368 (0.025)	1.42 (0.16)	25.4 (0.9)	0.105 (0.008)	22.3 (5.8)
	220	1573 (96)	1188 (99)	0.374 (0.016)	1.39 (0.14)	24.2 (0.7)	0.101 (0.008)	22.1 (2.3)
	240	1361 (92)	954 (99)	0.377 (0.029)	1.68 (0.10)	22.6 (1.0)	0.115 (0.007)	20.3 (3.4)
Block-PP	180	1455 (64)	1004 (91)	0.368 (0.010)	1.44 (0.18)	22.6 (0.7)	0.112 (0.006)	18.8 (4.4)
	200	1256 (71)	878 (97)	0.362 (0.015)	1.63 (0.05)	21.8 (1.4)	0.120 (0.009)	16.9 (2.9)
	220	1182 (52)	813 (45)	0.360 (0.015)	1.59 (0.05)	21.2 (1.3)	0.124 (0.005)	14.6 (1.5)
	240	1116 (23)	763 (42)	0.351 (0.010)	1.68 (0.04)	21.1 (0.6)	0.130 (0.005)	14.0 (1.3)

**Table 7 polymers-17-02107-t007:** Parameters used to ascertain critical expansion stress at different extrusion times.

Code	Extrusion Time [Cycles]	*E* [MPa]	*E*_υ_ [MPa]	υ [—]	ε_B_ [—]	σ_y_ [MPa]	*f*_0_ [—]	σ_v_ [MPa]
Homo-PP	0	1819 (89)	1261 (120)	0.345 (0.020)	1.33 (0.07)	27.6 (1.2)	0.105 (0.007)	21.7 (2.5)
	1	1740 (81)	1279 (106)	0.369 (0.023)	1.25 (0.07)	25.6 (2.2)	0.099 (0.008)	21.4 (3.1)
	3	1550 (60)	1027 (102)	0.385 (0.027	1.42 (0.26)	23.0 (1.3)	0.108 (0.007)	15.4 (6.8)
	5	1388 (40)	909 (50)	0.390 (0.034)	1.21 (0.09)	22.6 (1.0)	0.119 (0.006)	13.5 (3.2)
Block-PP	0	1305 (50)	910 (71)	0.361 (0.017)	1.63 (0.07)	22.2 (1.2)	0.118 (0.007)	17.7 (2.5)
	1	1241 (77)	888 (49)	0.370 (0.018)	1.67 (0.05)	21.4 (1.9)	0.116 (0.006)	18.0 (2.0)
	3	1190 (72)	780 (68)	0.350 (0.024)	1.66 (0.02)	21.4 (2.0)	0.129 (0.007)	14.3 (1.9)
	5	1046 (35)	662 (44)	0.341 (0.018)	1.70 (0.02)	20.3 (0.9)	0.141 (0.008)	11.3 (1.4)

**Table 8 polymers-17-02107-t008:** DSC results obtained at different extrusion temperatures.

Code	Extrusion Temp. [°C]	*T*_m_ [°C]	*X*_c_ [—]	*f* [—]
Homo-PP	180	163.8	0.465	0.0717
	200	163.2	0.475	0.0709
	220	163.2	0.473	0.0711
	240	164.3	0.465	0.0716
Block-PP	180	164.9	0.521	0.0666
	200	165.2	0.501	0.0684
	220	165.2	0.505	0.0680
	240	164.7	0.516	0.0671

**Table 9 polymers-17-02107-t009:** DSC results at different extrusion times.

Code	Extrusion Time [Cycles]	*T*_m_ [°C]	*X*_c_ [—]	*f* [—]
Homo-PP	0	164.1	0.483	0.0701
	1	164.0	0.474	0.0709
	3	164.2	0.461	0.0720
	5	164.0	0.452	0.0728
Block-PP	0	165.2	0.502	0.0683
	1	165.8	0.507	0.0677
	3	165.0	0.508	0.0678
	5	165.0	0.497	0.0687

## Data Availability

The data described as a result of this study are available on request from the corresponding author.

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
