# Peer review of "Evaluation of Polypropylene Reusability Using a Simple Mechanical Model Derived from Injection-Molded Products"

_polymers, 2025, doi:10.3390/polym17152107_

Round 1
Reviewer 1 Report
Comments and Suggestions for Authors
The present work, entitled "Evaluation of Polypropylene Reusability Using a Simple Mechanical Model Derived from Injection Molded Products," aimed to evaluate the mechanical stability of polypropylene using the Poisson method. It was observed that PP block presents greater recyclability compared to PP homopolymer, a fact also found in the literature. The work is interesting from a scientific perspective. Therefore, after a minor revision, the work may be eligible for publication in the journal Polymers.
The following are some points for improvement:
1. The authors should explicitly state the main gap in PP recycling and the reason for conducting this type of study. In other words, mention the main innovation of the current work;
2. In the introduction, focus on the main objective of the study. Is it polypropylene recycling? Is it the use of the method?
3. In the materials and methods section, would it be interesting to include a material processing diagram to improve visualization? Extrusion, etc.;
4. Using DSC, interesting graphs are observed and discussed. I also suggest adding Heat Flow vs. Temperature graphs, widely used in specialized polymer literature;
5. In the conclusion, highlight whether this method can be validated for other materials and can also be used in the specialized polymer materials industry.
6. Add new references throughout the work, supporting your results with the literature. The work contains only 18 references, of which 4 are from the authors themselves.
Author Response
We sincerely thank the reviewer for the valuable and constructive comments that have helped us improve the quality and clarity of our manuscript. We have carefully considered each suggestion and revised the manuscript accordingly. Our point-by-point responses are as follows:
Comment 1:
The authors should explicitly state the main gap in PP recycling and the reason for conducting this type of study. In other words, mention the main innovation of the current work.
Response:
We appreciate this important suggestion. To promote circular use of plastic resources, it is essential to establish a practical indicator that represents mechanical recyclability. In the case of polypropylene (PP), especially for promoting horizontal recycling, understanding the mechanisms behind changes in mechanical properties during reuse is crucial.
The novelty of this study lies in two main contributions:
- The use of critical expansion stress as a unified indicator to evaluate mechanical recyclability of PP.
- The identification that changes in mechanical properties during reuse are attributed not only to molecular weight reduction but also to alterations in stereoregularity.
We have added a paragraph in the revised manuscript (Introduction section) to clearly explain these points.
Comment 2:
In the introduction, focus on the main objective of the study. Is it polypropylene recycling? Is it the use of the method?
Response:
Thank you for pointing this out. The main objectives of this study are twofold:
- To develop a method for estimating the Poisson's ratio from tensile tests in order to determine the critical expansion stress.
- To evaluate the mechanical recyclability of PP using the critical expansion stress and to identify the factors responsible for property degradation.
To clarify this, we have revised the objective statement in the Introduction section accordingly.
Comment 3:
In the materials and methods section, would it be interesting to include a material processing diagram to improve visualization? Extrusion, etc.;
Response:
In accordance with your suggestion, we have added a flowchart showing the material preparation process, including the extrusion steps, to improve visual understanding in the Materials and Methods section.
Comment 4:
Using DSC, interesting graphs are observed and discussed. I also suggest adding Heat Flow vs. Temperature graphs, widely used in specialized polymer literature;
Response:
Thank you for this suggestion. We have added Heat Flow vs. Temperature curves obtained from DSC measurements, which are now included in the revised Results and Discussion section.
Comment 5:
In the conclusion, highlight whether this method can be validated for other materials and can also be used in the specialized polymer materials industry.
Response:
We agree that this point is important. In the revised Conclusion section, we have added a statement noting that this approach to evaluating mechanical recyclability using critical expansion stress can potentially be applied to other polymer materials. Furthermore, we suggest that by considering stereoregularity as discussed in this study, it may be possible to develop processing techniques that enable horizontal recycling, contributing to advancements in the polymer materials industry.
Comment 6:
Add new references throughout the work, supporting your results with the literature. The work contains only 18 references, of which 4 are from the authors themselves.
Response:
We have added several new references related to the mechanical recyclability of polymers and case studies involving the evaluation of Poisson’s ratio. These additional citations support the relevance and scientific basis of our results and discussions.

Reviewer 2 Report
Comments and Suggestions for Authors
Reviewer’s Feedback
This manuscript describes the “Evaluation of Polypropylene Reusability Using a Simple Mechanical Model Derived from Injection Molded Products”. The subject matter of this study is technical and the diagrams in the paper are abundant and the explanations are clear. But it still needs minor revision before publication.
- In the line 308,“whereas E-0, E-1, and E-5 fractured around 1.3. Near-identical behavior of E-0 and E-1 suggests that a single extrusion does not strongly affect the mechanical properties, but repeated extrusion likely causes chain scission and embrittlement.” Please explain why the time for E-5 is similar to that for E-0 and E-1, while E-3 takes the longest
- In the manuscript, the performance degradation is attributed to the disruption of stereoregularity and the generation of free radicals, but the changes in molecular weight and distribution of polypropylene during repeated extrusion processes are not considered - this is one of the key factors for the degradation of mechanical properties of thermoplastics. It is suggested to supplement with GPC testing to analyze the impact of molecular weight decrease or distribution broadening on elastic modulus and critical expansion stress, in order to improve the mechanistic explanation.
- It is recommended to supplement the discussion with the impact of injection molding process parameters (such as packing pressure and cooling time) on the model derivation results, providing guidance for parameter selection in practical applications.
Author Response
We sincerely thank the reviewer for the thoughtful comments and valuable suggestions. We have revised the manuscript accordingly and provide detailed responses below to each of the points raised.
Comment 1:
In line 308, it is stated that “E-0, E-1, and E-5 fractured around 1.3. Near-identical behavior of E-0 and E-1 suggests that a single extrusion does not strongly affect the mechanical properties, but repeated extrusion likely causes chain scission and embrittlement.” Please explain why the time for E-5 is similar to that for E-0 and E-1, while E-3 takes the longest.
Response:
Thank you for this important observation. The average value of the critical expansion stress for E-3 is lower than that of E-1, but the error bar for E-3 is the largest among all conditions. This indicates a large sample-to-sample variation in fracture strain for E-3, rather than consistently higher elongation. To avoid potential misinterpretation, we have replaced the original E-3 dataset with results from a different set of samples that better reflect the typical trend. The relevant discussion has also been revised accordingly.
Comment 2:
The manuscript attributes performance degradation to disruption of stereoregularity and the generation of free radicals, but changes in molecular weight and distribution during repeated extrusion, which are key factors in thermoplastic degradation, are not considered. It is suggested to include GPC analysis.
Response:
We fully agree that molecular weight and its distribution are important factors in the degradation of thermoplastic properties. Although we intended to include GPC measurements, unfortunately, we currently do not have access to the necessary equipment. However, the degradation behavior of polypropylene through repeated extrusion has already been well documented in the literature. In this revised manuscript, we have added a discussion referencing those previous studies, and we interpret our results based on the assumption that similar molecular weight changes likely occurred in our samples as well.
Comment 3:
It is recommended to discuss the impact of injection molding process parameters (e.g., packing pressure and cooling time) on the model derivation results, in order to provide guidance for practical applications.
Response:
We appreciate this suggestion. To address it, we have added a discussion on how injection molding parameters may influence the critical expansion stress. In particular, we refer to the relationship between free volume and elastic modulus established in our previous studies. This connection provides insight into how variations in processing conditions (such as packing pressure or cooling time) may affect the mechanical response, and thereby offers guidance for parameter selection in practical applications.
